# Dynamics of Systems with a Discontinuous Hysteresis Operator and Interval Translation Maps

Sergey Kryzhevich [1,*] , Viktor Avrutin [2], Nikita Begun [1], Dmitrii Rachinskii [3] and Khosro Tajbakhsh [4]

[1] Mathematics and Mechanics Faculty, Saint Petersburg State University, 199034 St. Petersburg, Russia; nikitabegun88@gmail.com

[2] Institute for Systems Theory and Automatic Control, University of Stuttgart, 70174 Stuttgart, Germany; viktor.avrutin@ist.uni-stuttgart.de

[3] Department of Mathematical Sciences, University of Texas at Dallas, Richardson, TX 75080, USA; dmitry.rachinskiy@utdallas.edu

[4] Faculty of Mathematical Sciences, Tarbiat Modares University, 14115-134 Tehran, Iran; khtajbakhsh@modares.ac.ir

[*] Correspondence: s.kryzhevich@spbu.ru

**Abstract:** We studied topological and metric properties of the so-called interval translation maps (ITMs). For these maps, we introduced the maximal invariant measure and study its properties. Further, we study how the invariant measures depend on the parameters of the system. These results were illustrated by a simple example or a risk management model where interval translation maps appear naturally.

**Keywords:** interval translation maps; ergodic measures; symbolic model; robustness; risk management

## 1. Introduction

Studying piecewise continuous maps, we can hardly rely on the classical numerical methods of approximating individual trajectories since the solutions are extremely sensitive to the variations of the initial conditions and the parameters of the system.

By contrast, invariant measures are more robust objects. As such, they provide a better language for representing the results of numerical models. In other words, if we model a discontinuous map, the picture on the computer screen represents invariant measures rather than topological attractors.

The paper consists of two parts. We start with a brief survey of some existing results of the ITM theory. Then, we describe the set of invariant measures for an interval translation map and discuss how they affect the topological properties of solutions. We introduce the so-called maximal invariant measure and study its properties. Another new result of this part of the paper is a theorem on the continuity of the set of invariant measures with respect to the residual set of parameters. This part of the paper extends and completes the results of [1].

In the second part of the paper, we consider a specific example—a model of a stock trader's behavior. We show that the associated map, for some generic set of its parameters, can be represented as an interval translation map. Finally, we study the parameter values for which the map is a rotation or a double rotation.

## 2. Interval Translation Maps: A Survey

Consider the unit segment $[0,1)$ endowed with the probability Lebesgue measure. Fix finite sets $\{t_k, k = 0, \ldots, n\}$ and $\{c_k, k = 1, \ldots, n\}$ such that:

$$0 = t_0 < t_1 < \ldots < t_n = 1, \qquad 0 \le t_{k-1} + c_k < t_k + c_k \le 1.$$

Let a piecewise continuous map $S : [0, 1) \to [0, 1)$ be defined by the formula $S(t) = t + c_k$ if $t \in I_k := [t_{k-1}, t_k)$. Then, $S$ is called an interval translation map (ITM). Such maps were first introduced by M. Boshernitzan and I. Kornfeld [2]. Replacing the segment $[0, 1)$ with the circle $\mathbb{T}^1$, one obtains a circle translation map (CTM).

To specify the number of segments, we also use the notation $n$-ITM and $n$-CTM for the interval and circle translation maps with $n$ segments, respectively.

If the images of the segments $I_k$ do not overlap, an ITM or a CTM is called an interval exchange map (IEM). Maps with flips, which include local maps of the form $S(t) = -t + c_k$, were also considered. However, in this paper, flips were by default excluded.

The basic properties of interval exchange maps were discussed and proven in [3], Section 14.5 (see also the surveys in [4,5]). However, the case of ITMs and CTMs seems to be more complicated.

**Definition 1.** *We say that an ITM S is finite if there exists a number $m \in \mathbb{N}$ such that $S^m([0, 1)) = S^k([0, 1))$ for any $k > m$. Otherwise, the map S is called infinite.*

In [2], the authors demonstrated that many ITMs are finite. As such, they can be reduced to interval exchange maps. However, there are examples with ergodic measures supported on Cantor sets. J. Schmeling and S. Troubetzkoy [6] provided estimates of the number of minimal subsets for such ITMs.

H. Bruin and S. Troubetzkoy [7,8] studied ITMs with three segments and demonstrated that in this case, a typical 3-ITM is finite. In the general case, they estimated the Hausdorff dimension of attractors and obtained sufficient conditions for the existence of a unique ergodic invariant measure. These results were generalized in [9] for ITMs with any number of segments. There is an uncountable set of parameters leading to infinite ITMs. However, the Lebesgue measure of these parameters is zero. Furthermore, some conditions ensuring the existence of multiple ergodic invariant measures were provided. H. Bruin and G. Clark [9] studied the so-called double rotations (2-CTMs); see also [10–12]. Almost all double rotations are finite. The parameters that correspond to infinite 2-CTMs form a set whose Hausdorff dimension is strictly between two and three.

J. Buzzi and P. Hubert [13] studied piecewise monotonous maps of zero entropy and no periodic points. In particular, they demonstrated that orientation-preserving ITMs without periodic points can have at most $n$ ergodic invariant probability measures where $n$ is the number of intervals.

D. Volk [14] demonstrated that almost every (w.r.t. the Lebesgue measure on the parameter set) 3-ITM is conjugated to either a rotation or a double rotation and is finite.

B. Pires in his preprint [15] proved that almost any ITM admits a non-atomic invariant measure (he assumed that the map does not have any connections or periodic points). This result was generalized by one of the co-authors in [1].

J. Buzzi [16] demonstrated that piecewise isometries defined on a finite union of polytopes have zero topological entropy in any dimension. In particular, this is true for interval translation maps.

A partition of the phase space induces a naturally defined symbolic dynamics. A. Goetz [17,18] demonstrated that for the case of the so-called regular partitions, the symbolic dynamics of an isometry cannot embed subshifts with positive entropy. He further obtained a condition for the polynomial growth of symbolic words. Under the assumptions of this paper, the partitions were also regular.

## 3. Invariant Measures

The following result was proven in [1] (see also [15]).

**Theorem 1.** *Any interval translation map S admits a Borel probability non-mixing invariant measure $\mu$.*

Now, let us observe that for any ITM or CTM, the set of corresponding invariant measures is closed in the space of all probability measures endowed with the $*$-weak topology. Indeed, a measure $\mu$ is invariant if and only if:

$$\int_0^1 \varphi(t)\, d\mu(t) = \sum_{k=0}^{n-1} \int_{t_k}^{t_{k+1}} \varphi(t + c_k)\, d\mu(t)$$

for any continuous function $\varphi : [0, 1) \to [0, 1)$. This equality holds true if we pass to the $*$-weak limit in the space of invariant measures. Therefore, similarly to [3], we can claim the existence of ergodic measures for ITMs. It was proven in [13] that any integral translation map admits at most $n$ non-atomic ergodic invariant measures.

**Definition 2.** *A non-empty segment (or interval) $J$ is called rigid with respect to an ITM $T$ if the restriction $T^m|_J$ is continuous for all $m \in \mathbb{N}$.*

It is well known that all points of a rigid segment are periodic.

Let $\{\mu_1, \ldots, \mu_k : k \leq n\}$ be the set of all ergodic probability non-atomic measures of the considered map (if any) and $\mu_{per}$ be the probability Lebesgue measure defined on the union of the rigid segments (if any); we call $\mu_{per}$ the periodic measure. We set either of these measures to zero if the corresponding set is empty. Let us define the maximal non-atomic measure:

$$\mu_{na} := \frac{1}{k} \sum_{j=1}^k \mu_j$$

and the maximal measure:

$$\mu^* := \frac{\mu_{na} + \mu_{per}}{2}$$

if the set of periodic points is non-empty. Otherwise, we define $\mu^* = \mu_{na}$.

Here, the wording "maximal" is justified by the following statement.

**Lemma 1.** *The support of any invariant measure is a subset of the support of the maximal measure. A similar statement is true for non-atomic measures.*

**Proof.** All non-atomic ergodic invariant measures are pairwise singular and, also, mutually singular with any periodic measure. As it was mentioned above, any ITM admits at most $n$ ergodic non-atomic invariant measures. Hence, any invariant measure of the studied map is a convex combination of a finite number of ergodic measures and a periodic measure, defined on the set of periodic points. Therefore, its support is a subset of the union of supports of the above measures. $\square$

We make a following conjecture. It appears correct for a big number of maps (including those with rational parameter). Numerical experiments show its validity, as well.

**Conjecture 1.** *Any ITM endowed with its maximal invariant measure is metrically conjugated to an IEM (perhaps of more than n segments). We also expect this conjugacy to be continuous almost everywhere.*

## 4. Minimal Points

First, we recall some basic definitions from topological dynamics.

**Definition 3.** *Let $f : K \mapsto K$ be a continuous map of a compact set $K$. A point $x_0 \in K$ is minimal if for any $y \in \overline{O^+}(x_0)$, we have $\overline{O^+(y_0)} = \overline{O^+(x_0)}$. A dynamical system is minimal if any point of the phase space is minimal.*

Here $O^+$ stands for the positive semi-orbit of a point.

**Definition 4.** *A set $A \subset \mathbb{N}$ is called syndetic if there exists an $N \in \mathbb{N}$ such that every block of $N$ consecutive positive integers intersects A.*

Given a dynamical system $(K, f)$, a point $x \in X$ is said to be syndetically recurrent if for every open neighborhood $U$ of $x$, the set of return times $\mathcal{N}(x, U) = \{n \in \mathbb{N} : f^n(x) \in U\}$ is syndetic.

The following lemma demonstrates a connection between syndetic recurrence and minimal systems for continuous dynamical systems.

**Lemma 2.** *Let $(X, f)$ be a continuous dynamical system on a compact metric space X.*

1.  *If $(X, f)$ is minimal, then every point $x \in X$ is syndetically recurrent.*
2.  *Conversely, if $x \in X$ is syndetically recurrent, then the closure of the orbit $O^+(x)$ is a minimal set.*

Although the above statement (as the vast majority of topological dynamics) is wrong for discontinuous systems, we can demonstrate that the points of the support of a maximal measure are "almost minimal".

**Lemma 3.** *Let $S$ be an interval translation map and $\mu^*$ be the maximal measure for this map. Then, for $\mu^*$—almost any point $x_0 \in \mathbb{T}^1$ and any open neighborhood $x_0 \in U$—the limit:*

$$F(U) := \lim_{n \to \infty} \frac{\#\{0 \le k \le n - 1 : f^k(x_0) \in U\}}{n}$$

*is well defined and positive.*

**Proof.** If a point $x_0$ is periodic, the above statement is evident. Otherwise, $x_0$ is a point of the support of one of the non-atomic ergodic invariant measures, call it $\mu_1$. By Birkhoff's ergodic theorem, the limit $F(U)$ exists for $\mu_1$—almost all points—and equals $\mu_1(U)$. $\square$

Given an interval translation map $S$, let $M(S)$ be the support of its maximal measure.

**Conjecture 2.** *Almost all points of the set $M(S)$ are syndetically recurrent.*

In order to be able to discuss the topological minimality of the points of the set $M(S)$, we introduce a symbolic model for our dynamics.

Given an $n$-ITM $S$, we define the set $P(S)$ as the closure of all periodic points of $S$ or, in other words, the closure of all rigid segments. Now, we introduce the set $\Omega$ of all one-sided sequences of $n$ symbols $1, \ldots, n$, endowed with the standard topology.

Let $[0, 1] = I_1 \bigcup \ldots \bigcup I_n$ be the partition that corresponds to the map $S$.

Consider:

$$\Omega_S := \{\omega = \{\omega_m, m \in \mathbb{Z}^+\} : \exists x \in [0, 1] : S^m(x) \in J_{\omega_m}\}.$$

**Definition 5.** *An orbit of a discontinuity point $t_i$ is called a separatrix if there exist $k \in \mathbb{N}$ and $j \in \{0, \ldots, n\}$ such that $S^k(t_i) = t_j$ or $S^k(t_i - 0) = t_j$.*

**Lemma 4.** *Let ITM $S$ have no separatrices and is conjugated to an IEM. Then, the symbolic model $\Omega_S$ endowed with the shift map is minimal.*

**Proof.** The statement of the lemma is true for interval exchange maps; see [3], Corollary 14.5.12 and Exercise 14.5.2. Then, we can apply our conjugacy assumption and thus prove the lemma. $\square$

As the numerical results show (see Section 9), the set $M(S)$ appears to coincide with the set:

$$\Xi = \bigcup_{N=1}^{\infty} \overline{S^n(\mathbb{T}^1)}.$$

**Conjecture 3.** $\Xi = M(S)$.

Since all points of $\Xi$ are non-wandering, the validity of the above conjecture would imply that all non-wandering points of an interval translation map are minimal (the converse statement is trivial).

### 5. Convergence of Invariant Measures

In this section, we provide a result that justifies numerical modeling. Any numerical methods can give us a set of periodic measures distributed on rigid segments for ITMs with rational parameters. It is interesting if these measures provide a good approximation to an invariant measure of a non-periodic ITM.

This can be illustrated by the following theorem.

**Theorem 2.** *Let $S_m$ ($m \in \mathbb{N}$) and $S_*$ be ITMs of n intervals, and let the parameters of $S_m$ converge to those of $S_*$. Let $\mu_m$ be an invariant probability measure for $S_m$, and suppose that the measures $\mu_m$ converge $*$-weakly to a measure $\mu_*$. Then, $\mu_*$ is an invariant measure for $S_*$ provided that $\mu_*\{t_k^*\} = 0$ for any k, where $t_k^*$ are discontinuity points of the map $S_*$.*

**Proof.** Let $t_k^m$ and $c_k^m$ be the parameters of maps $S_m$ and $t_k^*, c_k^*$ be those of the map $S_*$. Given a continuous map $\varphi : [0,1] \to [0,1]$, we have:

$$\int_0^1 \varphi(t) \, d\mu_m(t) = \sum_{k=0}^{n-1} \int_{t_k^m}^{t_{k+1}^m} \varphi(t + c_k^m) \, d\mu_m(t) \tag{1}$$

for all $m \in \mathbb{N}$. The principal question is whether or not we can pass to the $*$-weak limit in Equation (1).

It goes without saying that the left-hand side of (1) converges to $\int_0^1 \varphi(t) \, d\mu_*(t)$; we need to check whether the right-hand side converges to:

$$\sum_{k=0}^{n-1} \int_{t_k^*}^{t_{k+1}^*} \varphi(t + c_k^*) \, d\mu_*(t)$$

Let $M = \max_{t \in [0,1]} \varphi(t)$. Given a $\varepsilon > 0$, we take an $m \in \mathbb{N}$ so large that $|t_k^m - t_k^*| < \varepsilon$,

$$\left| \int_{t_k^* + \varepsilon}^{t_{k+1}^* - \varepsilon} \varphi(t + c_k^*) \, d\mu_m - \int_{t_k^* + \varepsilon}^{t_{k+1}^* - \varepsilon} \varphi(t + c_k^*) \, d\mu_* \right| < \varepsilon,$$

and:

$$|\varphi(t + c_k^m) - \varphi(t + c_k^*)| < \varepsilon$$

for all $k \in \{0, \dots, n-1\}$ and all $t \in [0,1]$ for which the latter inequality is well defined. Then, for any $k \in \{0, \dots, n-1\}$, we have:

$$
\left| \int_{t_k^*}^{t_{k+1}^*} \varphi(t + c_k^*) \, d\mu_*(t) - \int_{t_k^m}^{t_{k+1}^m} \varphi(t + c_k^*) \, d\mu_*(t) \right|
$$

$$
\leq \left( \int_{t_k^* - \varepsilon}^{t_k^* + \varepsilon} + \int_{t_{k+1}^* - \varepsilon}^{t_{k+1}^* + \varepsilon} \right) M \, d\mu_*(t) + \left| \int_{t_k^* + \varepsilon}^{t_{k+1}^* - \varepsilon} \varphi(t + c_k^*) \, d\mu_m - \int_{t_k^* + \varepsilon}^{t_{k+1}^* - \varepsilon} \varphi(t + c_k^*) \, d\mu_* \right|
$$

$$
+ \int_{t_k^* + \varepsilon}^{t_{k+1}^* - \varepsilon} |\varphi(t + c_k^*) - \varphi(t + c_k^m)| \, d\mu_m(t) \leq M\mu_* + 2\varepsilon.
$$

This proves the theorem. □

## 6. A Model of a Trader's Behavior

The rest of the paper is devoted to an example of interval translation maps that appears in modeling real-life problems, namely in a simple risk management model.

The main ingredient of this model is the so-called stop operator, first introduced in [19]; see also [20–22] for surveys on the problem.

Various models of real-life problems were modeled using the stop operator techniques, and we mention those related to economics: [23–25].

Going back to our model, we considered a trader with a strategy that depends on some economic parameters. At every step (for instance, once a month), the trader varies the strategy based on the current values of the parameters. In reality, it is often more convenient to disregard small changes of parameters taking into account the cost of actions (for instance, the forced loss of money in currency exchange). However, as soon as the risk reaches the critical value, the parameters of the strategy are suddenly changed so that the strategy becomes optimal.

A very similar model was developed in [26]. However, in [26], the parameters were kept at the critical level. Although the corresponding strategy results in continuous dynamics, the discontinuous optimization approach considered below looks more relevant in many real-life situations. As we will see, this difference in assumptions changes the dynamics dramatically. In particular, the so-called saw map considered in [26,27] demonstrates a chaotic behavior conjugated to complete symbolic dynamics (Bernoulli shift), while the discontinuous model described below does not. On the other hand, the dynamics we obtained has a significant similarity with the sociological model of opinion dynamics studied by Pilyugin and Sabirova [28].

## 7. Model Equations

Let $s_0 \in [-1, 1]$, and let $\{x_n\}$, $n \in \mathbb{N}_0$, be a real-valued sequence. The operator $\Sigma$ maps a pair $(s_0, \{x_n, n \in \mathbb{N}_0\})$ to the sequence defined by the formula:

$$
s_{n+1} = \Psi(s_n + x_{n+1} - x_n), \quad n \in \mathbb{N}_0,
$$

where the function $\Psi$ is the cut-off function for the interval $[-1, 1]$:

$$
\Psi(\tau) = \begin{cases} 0 & \text{if} \quad |\tau| > 1, \\ \tau & \text{if} \quad |\tau| \leq 1. \end{cases}
$$

Below, we call the operator $\Sigma$ the risk operator. Denote:

$$
\mathcal{L} = \{(x, s) : x \in \mathbb{R}, s \in [-1, 1]\}.
$$

We fix $(x_0, s_0) \in \mathcal{L}$ and consider the dynamical system:

$$\begin{cases} x_{n+1} = \lambda x_n + \alpha s_n, \\ s_{n+1} = \Psi(s_n + x_{n+1} - x_n), \end{cases} \tag{2}$$

with $\lambda \in (-1, 1)$, $\alpha \in \mathbb{R}$, $n \in \mathbb{N}_0$. It is easy to see that $(x_n, s_n) \in \mathcal{L}$, $n \in \mathbb{N}_0$. Thus, the strip $\mathcal{L}$ is the phase space for System (2). For the sake of convenience, we write System (2) as follows:

$$\begin{pmatrix} x_{n+1} \\ s_{n+1} \end{pmatrix} = f\begin{pmatrix} x_n \\ s_n \end{pmatrix}.$$

Our objective is to understand the dynamics of System (2) for various values of parameters $\lambda$ and $\beta = \lambda + \alpha$. First, let us observe two simple properties of the map $f$.

**Lemma 5.** *All trajectories of the system* (2) *are bounded. Fixed points of System* (2) *form the segment:*

$$EF = \left\{ (x, s) : \ x = \frac{\alpha s}{1 - \lambda}, \ -1 \le s \le 1 \right\}$$

*with the end points:*

$$E = \left( \frac{\alpha}{1 - \lambda}, 1 \right), \quad F = \left( -\frac{\alpha}{1 - \lambda}, -1 \right).$$

The proof of this statement is trivial.

Below, we use the standard notion of stability and instability for equilibria. We also say that an equilibrium point $(x^*, s^*)$ of System (1) is semi-stable if there are open sets $U_1, U_2 \subset \{(x, s) : |s| < 1\}$ such that $(x^*, s^*)$ belongs to their boundaries and, simultaneously:

- for every $\varepsilon > 0$, there is a $\delta > 0$ such that any trajectory starting from the $\delta$-neighborhood of the equilibrium point $(x^*, s^*)$ in the set $U_1$ belongs to the $\varepsilon$-neighborhood of $(x^*, s^*)$ for all positive $n$;
- there is an $\varepsilon_0 > 0$ such that any trajectory starting in $U_2$ leaves the $\varepsilon_0$-neighborhood of the equilibrium $(x^*, s^*)$ after a finite number of iterations.

**Lemma 6.** *For any $\beta \in (-1, 1)$, all equilibrium points except E and F are stable, while points E and F are semi-stable.*

The proof of this statement is also trivial, and we leave it to the reader.

Throughout the proofs of the following statements, we denote by $A_x$ and $A_s$ the $x$ and $s$ coordinates of a point $A \in \mathcal{L}$, respectively; a similar notation is used for other points. We also use the variable $p = x - s$. Due to the symmetry $f(-x, -s) = -f(x, s)$, it suffices to present the proofs for the right half space.

**Lemma 7.** *For any point $A$ to the left of the segment $EF$, one has $[f(A)]_x > A_x$. For any point $B$ to the right of the segment $EF$, one has $[f(B)]_x < B_x$.*

**Proof.** Since $A$ lies to the left of the segment $EF$, one has $(1 - \lambda)A_x < \alpha A_s$. Hence,

$$[f(A)]_x = \lambda A_x + \alpha A_s > \lambda A_x + (1 - \lambda)A_x = A_x.$$

A similar argument applies to the point $B$. □

Denote by $\Pi \subset \mathcal{L}$ the parallelogram with the diagonal $EF$, two sides on the lines $s = \pm 1$, and two sides with slope one(see Figure 1):

$$\Pi = \left\{ (x, s) : \ |x - s| \le \left| \frac{\alpha}{1 - \lambda} - 1 \right|, \ |s| \le 1 \right\}.$$

**Lemma 8.** *Let $(x_0, s_0) \in \Pi$ and $p_0 = x_0 - s_0$. If either $-1 < \beta < 0$ and $|\beta x_0 - (\alpha+1)p_0| \leq 1$ or $0 \leq \beta < 1$, then the trajectory with the initial point $(x_0, s_0)$ satisfies $p_n = x_n - s_n = p_0$ for all $n \geq 0$ and $(x_n, s_n) \to (x_*, s_*)$ as $n \to \infty$, where:*

$$(x_*, s_*) = \left( -\frac{\alpha p_0}{1-\beta}, -\frac{(1-\lambda)p_0}{1-\beta} \right) \tag{3}$$

*is a fixed point of $f$ with $x_* - s_* = p_0$.*

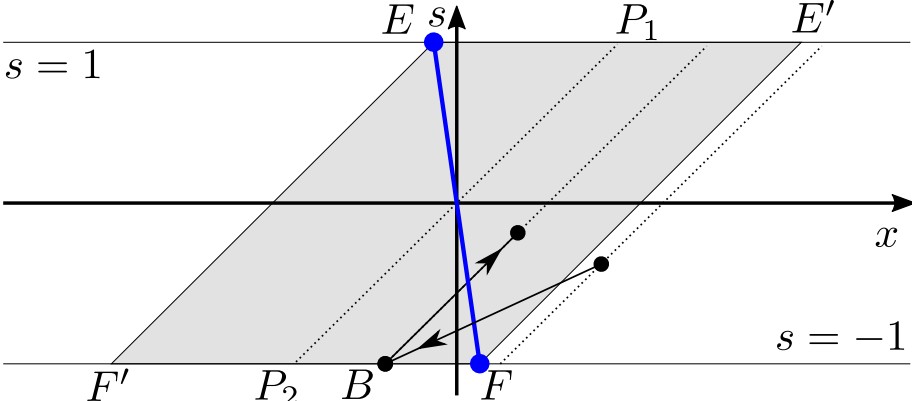

**Figure 1.** Parallelogram $\Pi$.

**Proof.** Consider the sequence $(x_n, s_n)$ defined by:

$$x_{n+1} = \beta x_n + \alpha(s_n - x_n), \qquad s_{n+1} = x_{n+1} - x_n + s_n \tag{4}$$

and suppose that $|s_n| \leq 1$ for all $n$. Then, this sequence is a trajectory of (1). Equation (4) is equivalent to the relations $x_n - s_n = p_0$, $x_{n+1} = \beta x_n - \alpha p_0$, which result in the explicit formulas:

$$x_n - x_* = s_n - s_* = \beta^n(x_0 - x_*) = \beta^n(s_0 - s_*), \tag{5}$$

where we use the notation (3). Since $s_0, s_* \in [-1, 1]$, Equation (5) implies $|s_n| \leq 1$ for all $n$ if $0 \leq \beta < 1$. This means that the trajectory converges to the equilibria $(x_*, s_*)$ along the slanting segment $p = x - s = p_0$ and never leaves the parallelogram $\Pi$. Similarly, if $-1 < \beta < 0$ and, in addition, $|s_* + \beta(s_0 - s_*)| \leq 1$, then Equation (6) also implies $|s_n| \leq 1$ for all $n$. By definition of $s_*$, we have $s_* + \beta(s_0 - s_*) = \beta x_0 - (\alpha+1)p_0$; hence under the assumptions of the lemma, the sequence $s_n$ satisfies $|s_n| \leq 1$, and therefore, the formulas explicitly define a trajectory of (2). This trajectory also converges to the equilibrium $(x_*, s_*)$ "jumping" over it along the slanting segment $p = p_0$ and never leaving the parallelogram $\Pi$. $\square$

Lemma 8 does not describe the destiny of those trajectories that either start from outside of the parallelogram $\Pi$ or have an initial point $(x_0, s_0) \in \Pi$ such that:

$$|\beta x_0 - (\alpha+1)p_0| > 1, \qquad -1 < \beta < 0.$$

Note that for all of those trajectories, there exists $k \in \mathbb{N}$ such that $|s_k + x_{k+1} - x_k| > 1$ (i.e., these trajectories "feel" the discontinuity of System (2)). Below, we show how in this case, we can reduce System (2) to a one-dimensional Poincaré map.

In this paper, we analyzed the case $\beta > 1$, $\lambda < 0$ only. Other cases could be considered similarly.

**Theorem 3.** *Let $\lambda \in (-1, 0)$, $\beta > 1$. Then:*

1.  *For any point $(x_0, s_0) \notin EF$, there is a number $k \in \mathbb{N}$ such that:*

$$f^k(x_0, s_0) \in \{(x, 0)\}.$$

2.  *There exists a non-atomic invariant measure $\mu$ of the map $f$ such that $\text{supp}\,\mu \cap EF = \emptyset$.*
3.  *The set of non-wandering points of the map $f$ that do not belong to the equilibrium set $EF$ is uncountable.*

## 8. Proof of Theorem 3

*8.1. Reduction to a One-Dimensional Map*

For $\beta > 1$, all equilibrium points are unstable. Hence, if $(x_0, s_0) \notin EF$, then there exists a minimal $i = i(x_0, s_0) \in \mathbb{N}$ such that $s_i = 0$. Here, $(x_i, s_i)$ is the $i$-th iteration of the point $(x_0, s_0)$. Without loss of generality, we assume that $s_0 = 0$ and denote $i(x_0) = i(x_0, 0)$.

Consider the Poincaré map $\hat{T} : \mathbb{R} \to \mathbb{R}$ defined as $\hat{T}(x) = x_{i(x)}$, $\hat{T}(0) = 0$. Since the map $f$ is odd, so is the map $\hat{T}$. From $\lambda < 0$, it follows that $x_1 = \lambda x_0 < 0$ for $x_0 > 0$, and hence, $\hat{T}(x) < 0$, $x > 0$. Define $T(x) : [0, +\infty) \mapsto [0, +\infty)$ as $T(x) = |\hat{T}(x)|$, $x \geq 0$. By definition,

$$
\begin{aligned}
\hat{T}(x) &= -T(x), & x &\geq 0, \\
\hat{T}(x) &= T(-x), & x &< 0, \\
\hat{T}^2(x) &= T^2(x), & x &\geq 0.
\end{aligned}
\tag{6}
$$

It is easy to see that the dynamics of $f$ and $T$ are closely connected. In particular, each periodic point $x$ of the map $T$ corresponds to the periodic point $(x, 0)$ of the map $f$. The equalities (6) allow us to restrict the analysis to positive values of $x$ only.

*8.2. Structure of the Map T*

First of all, note that as long as $|s_n + x_{n+1} - x_n| \leq 1$, System (2) can be rewritten in the following way:

$$
\begin{pmatrix} x_{n+1} \\ s_{n+1} \end{pmatrix} = A \begin{pmatrix} x_n \\ s_n \end{pmatrix},
$$

where $A$ is the $2 \times 2$ matrix defined by:

$$
A = \begin{pmatrix} \lambda & \alpha \\ \lambda - 1 & \alpha + 1 \end{pmatrix}.
\tag{7}
$$

Let us show that for any $k \in \mathbb{N}$, there exists a unique point $Q_k = (q_k, 0)$, $q_k > 0$ such that first $k - 1$ iterations of the point $(q_k, 0)$ under the map $f$ belong to the interior of $\mathcal{L}$, and the $k$-th iteration is $(q_k - 1, -1)$. Indeed, it follows from (7) that:

$$
A^n = \frac{1}{\beta - 1} \begin{pmatrix} \alpha - (1 - \lambda)\beta^n & \alpha\beta^n - \alpha \\ (\lambda - 1)(\beta^n - 1) & \alpha\beta^n - 1 + \lambda \end{pmatrix}
\tag{8}
$$

and by the definition of $q_k$, we have:

$$
A^k \begin{pmatrix} q_k \\ 0 \end{pmatrix} = \begin{pmatrix} q_k - 1 \\ -1 \end{pmatrix}.
\tag{9}
$$

From (8) and (9), we obtain the explicit formula for $q_k$:

$$
q_k = \frac{\beta - 1}{(1 - \lambda)(\beta^k - 1)}.
$$

It is also easy to obtain the following statements (see Figure 2):

- $T$ is discontinuous at the points $q_k$, $k \in \mathbb{N}$;
- $T$ is continuous on the sets $(q_1, +\infty)$ and $(q_{k+1}, q_k]$, $k \in \mathbb{N}$;
- $\lim_{k \mapsto \infty} q_k = 0$;

- $T(q_k) = -(\lambda(q_k - 1) - \alpha) = \beta - \lambda q_k, k \in \mathbb{N}$;
- $T(q_k) > T(q_{k+1}), k \in \mathbb{N}$;
- $\lim_{k \mapsto \infty} T(q_k) = \beta$;
- $\lim_{x \mapsto q_k + 0} T(x) = 1 - q_k, k \in \mathbb{N}$;
- $\lim_{x \mapsto q_k + 0} T(x) < \lim_{x \mapsto q_{k+1} + 0} T(x), k \in \mathbb{N}$;
- $\lim_{k \mapsto +\infty} \lim_{x \mapsto q_k + 0} T(x) = 1$;
- on the segment $(q_{k+1}, q_k], k \in \mathbb{N}$, the map $T(x)$ is given by the equation:

$$T(x) = \chi_k x,$$

where:

$$\chi_k := \frac{T(q_k)}{q_k} > 0;$$

- on the segment $(q_1, +\infty)$, the map $T(x)$ is given by the equation $T(x) = -\lambda x$.

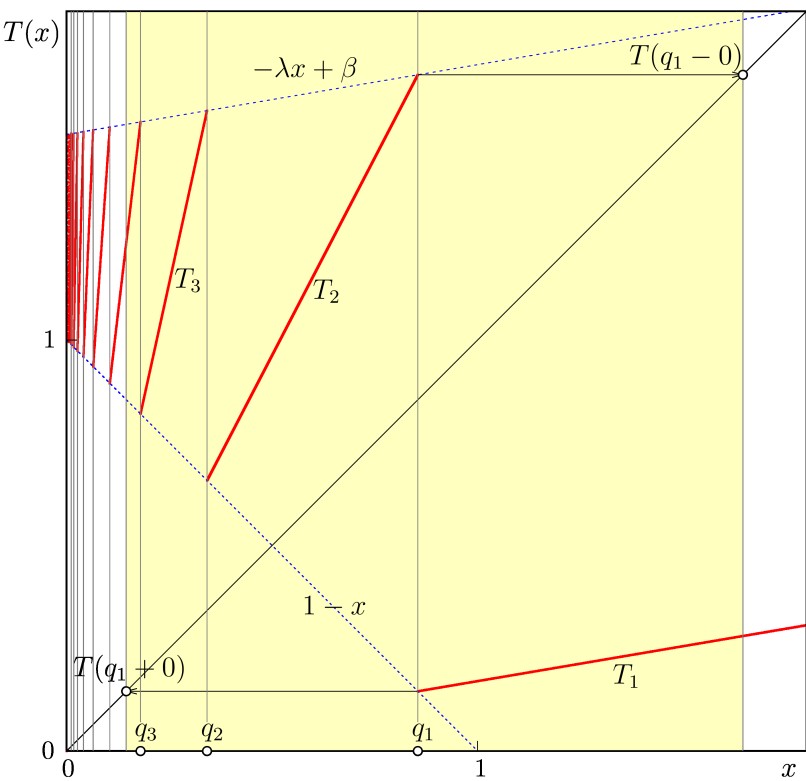

**Figure 2.** A sample form of the map $T$.

Denote:

$$\mu = \lim_{x \mapsto q_1 + 0} T(x) = -\frac{\lambda}{1 - \lambda},$$

$$\nu = T(q_1) = \beta - \frac{\lambda}{1 - \lambda}.$$

Taking into account the above statements, the following lemmas are straightforward.

**Lemma 9.** *The segment $[\mu, \nu]$ is positively invariant under the map $T$.*

**Lemma 10.** *The Lyapunov exponents:*

$$\lim_{n \to +\infty} \frac{1}{n} \ln |DT^n(x)|$$

*exist and are equal to zero for almost all $x \in \mathbb{R}$.*

As stated above, the map $T$ is piecewise linear, and $T(x) = \chi_k x$ on every segment $(q_{k+1}, q_k]$ with $\chi_k > 0$. Let us apply the transformation of the variable $y = \ln x$, $x \in (0, \infty)$. In terms of the variable $y$, the map $T$ defined on $[\mu, \nu]$ transforms to the map $S(y) = y + b_k := y + \ln \chi_k$ defined on the segment $[\ln \mu, \ln \nu]$. Maps $T$ and $S$ are obviously topologically conjugated. We see that $S$ is a classic interval translation map with no flips.

### 8.3. Interval Translations of Two and Three Intervals

Note that if $\mu > q_2$, then the point $q_1$ is the only discontinuity point of the map $T$ inside $[\mu, \nu]$. Furthermore, the inequality $\mu > q_2$ is equivalent to the inequality $-\lambda(\beta + 1) > 1$.

Assume that $\mu > q_2$, after identifying the points $\ln \mu$ and $\ln \nu$. In this case, the map $S$ becomes a rigid rotation by the angle $\vartheta = \ln(\chi_2/\mu)$ of the circle $\mathbb{T}^1$, which has the length $\theta = \ln(\nu/\mu)$. As such, if the rotation number $\vartheta/\theta$ is rational, then each trajectory is periodic; on the other hand, if the rotation number $\vartheta/\theta$ is irrational, then each trajectory is dense in $\mathbb{T}^1$.

In the case $-\lambda(\beta + 1) \leq 1 < -\lambda(\beta^2 + \beta + 1)$, the same procedure can be used to identify $S$ with a double rotation (see Section 2). In general, $S$ is equivalent to an $n$-CTM if:

$$\frac{-\lambda(\beta^n - 1)}{\beta - 1} \leq 1 < \frac{-\lambda(\beta^{n+1} - 1)}{\beta - 1}.$$

## 9. Numerical Simulations

Figure 3a presents the results of numerical simulations of System (2) for the fixed value $\beta = 1.5$ with $\alpha$ varying from one to 1.42. Note that in contrast to our previous assumptions, the parameter $\lambda$ is positive.

Overall, for the most parameter values, the considered ITM appears finite, and the support of the maximal measure consists of a finite number of intervals. However, these intervals can shrink to points under parameter variation. The dependence on $\alpha$ appears to be continuous for almost all values of the parameter. On the other hand, there are many (probably, infinitely many) discontinuity points.

The boundaries of the intervals constituting the set $M(S)$ depend linearly on $\alpha$. This can be explained by the fact that the boundaries of these intervals are given by the images of the limiting values of $T(q_k)$. It can easily be shown that the set $M(S)$ is just one interval for $\alpha \geq \alpha^*$ with $\alpha^* = \frac{\beta+2}{\beta+1}$. Indeed, in this case, the map on its absorbing interval has a single discontinuity $q_1$, and the condition $\alpha = \alpha*$ corresponds to the case $T(q_1 + 0) = q_2$, i.e., the discontinuity $q_2$ located at the boundary on the absorbing interval.

The organizing principles of the presented bifurcation structure are still to be investigated in detail; however, it is worth noticing a striking similarity between this structure and the band count adding structure known for multi-band chaotic attractors. In the latter case, as initially reported in [29] (for more details, we refer to [30]), between two parameter intervals associated with the $\mathcal{K}_1$-band and $\mathcal{K}_2$-band chaotic attractors, there is a parameter interval associated with the $(\mathcal{K}_1 + \mathcal{K}_2 - 1)$-band chaotic attractors. The same regularity can be observed in System (2). For example, in Figure 3b, one can see parameter intervals where the sets $M(S)$ consist of three and four bands (connected components), a parameter interval corresponding to sets $M(S)$ consisting of six bands in between, and so on. However, there is a major difference in the case described in the literature, as the band count adding structure has been so far reported for chaotic attractors only, while the sets $M(S)$ in System (2) are not chaotic. To the best of our knowledge, a band count adding structure formed by non-chaotic attractors has never been reported before. To which extent the bifurcation scenario observed in System (2) follows the regularities of the band count adding structure is still to be investigated.

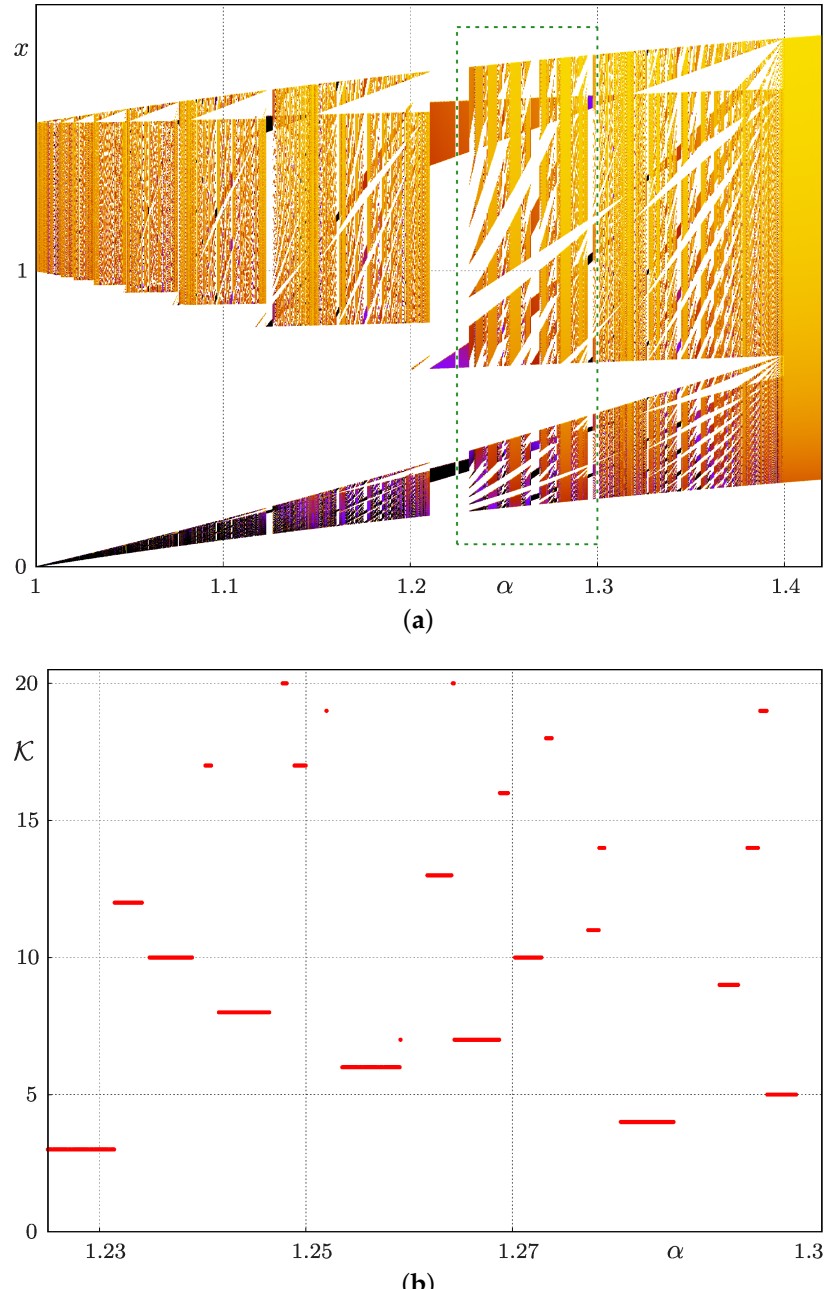

**Figure 3.** (**a**) The set $M(S)$ for various values of parameters $\alpha$ and $\beta = 1.5$. (**b**) The number of bands (connected components) of sets $M(S)$ for parameter values indicated by the rectangle marked in (**a**).

## 10. Discussion

Based on the principal idea of [1], we demonstrated that there is a similarity between interval translations and interval exchange maps. We showed the continuity of invariant measures with respect to the parameters of the system.

As an example, we discussed the dynamics of a simple risk management model, studied the properties of this map, and simulated it numerically. These simulations were in complete agreement with the theoretical predictions, including those of this paper.

**Author Contributions:** Theoretical part, S.K.; tasking and methodology, D.R.; numerical simulations, V.A.; studying a risk management model, N.B. and K.T. All authors read and agreed to the published version of the manuscript.

**Funding:** This research received no external funding.



**Acknowledgments:** Viktor Avrutin was supported by DFG, AV 111/2-2. Nikita Begun was supported by the Russian Foundation for Basic Researches, Grants 19-01-00388 and 18-01-00230 (Sergey Kryzhevich was also funded from the latter grant). Nikita Begun and Sergey Kryzhevich are grateful to B. Fiedler and his research group at Free University of Berlin for arranging excellent conditions to meet and to work on this project. All authors are grateful to S. Troubetskoy for his remarks and comments that made it possible to corrects some mistakes in the initial version of the text.

**Conflicts of Interest:** The authors declare no conflict of interests.

## Abbreviations

The following abbreviations are used in this manuscript:

| IEM | Interval exchange map |
|-----|-----------------------|
| ITM | Interval translation map |
| CTM | Circle translation map |

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
