# Peer review of "Dynamics of Systems with a Discontinuous Hysteresis Operator and Interval Translation Maps"

_axioms, doi:10.3390/axioms10020080_

Round 1
Reviewer 1 Report
This paper represents a nice contribution to the theory of interval translation maps. The authors introduce the so-called maximal invariant measure associated to an invariant translation map and show that with such measure, the map is metrically conjugated to an interval exchange map. This enables authors to extend some known properties of interval translation maps.
I find the paper to be correct, interesting and novel. Therefore, I recommend its acceptance in Axioms.
Author Response
We cordially thank the Reviewer for his/her work and positive opinion.
Reviewer 2 Report
In this paper, it is demonstrated that there is a significant similarity between interval translations and interval exchange maps. It is illustrated with various statements that are novel for interval and circle translations. In particular, a canonical metric equivalence may be introduced. It is studied the continuity of invariant measures with respect to parameters of the system and, finally, it is proved a ’toy version of the Closing Lemma. Thereafter, it is considered a simple risk management model, studied the properties of this map and performed numerical studies that are in complete agreement with the theoretical results including those of this paper.
The paper contains interesting, new, and nontrivial results.
Hence it deserves to be published.
Comments:
- 119+120(also 144) what is the difference of "\rightarrow" and "\mapto" in all text
- 142+196 delete "dot"
- 148 what is the connection of synd.rec. to (uniform) recurrence?
- 158 in Lm5 better $x_0 \in U$
- quality of Fig.2 is not acceptable
- 319+320 pls add ref to the investigated model
- Fig.1 the horizontal axes stands for "a" or "\alpha"?
- A list of references connected to the introduction part can be improved
Author Response
We are grateful to the Reviewer for his/her work, valuable opinions and, of course, for the helpful comments and remarks
We took them into account and made the following changes.
0.Moderate English changes required
The grammar was carefully checked throughout the text.
1. 119+120(also 144) what is the difference of "\rightarrow" and "\mapto" in all text.
There was no difference. This occurred because the text was prepared by several co-authors simultaneously. Anyway, now we have ‘\mapsto’ only.
2. 142+196 delete "dot"
Done.
3. 148 what is the connection of synd.rec. to (uniform) recurrence?
We added the phrase ‘The following lemma demonstrates a connection between syndetic recurrence and minimal systems for continuous dynamical systems.’ which is followed by a classical lemma from Continuous Dynamics (line 148).
4. 158 in Lm5 better $x_0 \in U$
Done.
5. quality of Fig.2 is not acceptable
A better version of Fig.2 is uploaded.
6. 319+320 pls add ref to the investigated model
We are not sure, the investigated model has ever been studied before (please, correct us if we are wrong!). However, it seems, our model is a natural modification of the classical stop operator that appears in a wide range of applications.
The reference [7] (a close model studied by one of co-authors) was added as well as some other references concerning stop operator theory.
7. Fig.1 the horizontal axes stands for "a" or "\alpha"?
Probably, the question concerns Fig 3a (former Fig 3). The figure was updated and corrected.
8. A list of references connected to the introduction part can be improved
We added a couple of classical works related to Interval Exchange Maps, and three references about double rotations. However, we are not sure, this is what the Reviewer requests us to do. If there exists any specific reference, we need to add, we kindly ask him/her a favor to let us know what is it.
Please see the attachment.
